# Replacement of the Genomic Scaffold Improves the Replication Efficiency of Synthetic *Klebsiella* Phages

**DOI:** 10.3390/ijms26146824

**Published:** 2025-07-16

**Authors:** Ivan K. Baykov, Olga M. Kurchenko, Ekaterina E. Mikhaylova, Anna V. Miroshnikova, Vera V. Morozova, Marianna I. Khlebnikova, Artem Yu. Tikunov, Yuliya N. Kozlova, Nina V. Tikunova

**Affiliations:** 1Laboratory of Molecular Microbiology, Institute of Chemical Biology and Fundamental Medicine of the Siberian Branch of the Russian Academy of Sciences, Novosibirsk 630090, Russiamorozova@niboch.nsc.ru (V.V.M.);; 2Shared Research Facility “Siberian Circular Photon Source” (SRF “SKIF”), Boreskov Institute of Catalysis of the Siberian Branch of the Russian Academy of Sciences, Novosibirsk 630090, Russia; 3Faculty of Natural Sciences, Novosibirsk State University, Novosibirsk 630090, Russia

**Keywords:** bacteriophage, *Klebsiella*, tailspike depolymerase, receptor-binding protein, synthetic biology, genome assembly, transformation-associated recombination cloning

## Abstract

In this study, the impact of the genomic scaffold on the properties of bacteriophages was investigated by swapping the genomic scaffolds surrounding the tailspike genes between two *Przondovirus* phages, KP192 and KP195, which infect *Klebsiella pneumoniae* with different capsular types. A yeast-based transformation-associated recombination cloning technique and subsequent “rebooting” of synthetic phage genomes in bacteria were used to construct the phages. Using *Klebsiella* strains with K2, K64, and KL111 capsular types, it was shown that the capsular specificity of the synthetic phages is fully consistent with that of the tailspike proteins (*tsp*). However, the efficiency of plating and the lytic efficiency of these phages strongly depended on the genomic scaffold used and the *Klebsiella* strain used. Synthetic phages with swapped genomic scaffolds demonstrated superior reproduction efficiency using a number of strains compared to wild-type phages, indicating that some elements of the swapped genomic scaffold enhance phage replication efficiency, presumably by blocking some of the host anti-phage defense systems. Our findings demonstrate that even in the case of closely related phages, the selection of the genomic scaffold used for *tsp* gene transplantation can have a profound impact on the efficiency of phage propagation on target bacterial strains.

## 1. Introduction

Phage therapy is one of the promising approaches to control infections caused by antibiotic-resistant bacteria [1,2]. This approach is especially relevant due to the wide spread of multidrug-resistant bacterial strains. However, natural phages can have low lytic activity against the target infectious strain. The design of synthetic phages based on natural bacteriophages opens up new possibilities. A targeted combination of genetic elements can lead to the control of specificity and other characteristics of the phage (reviewed in [3,4,5,6,7,8]).

The success of the rational design of therapeutic phages depends on several factors. First, this requires effective methods of editing and assembling phage genomes. Currently, a number of such methods have been developed, including transformation-associated recombination (TAR) cloning in yeast (also known as gap repair cloning) [9,10], the assembly of phage genomes using the Gibson method [11], and various recombination methods inside bacterial host cells like Bacteriophage Recombineering of Electroporated DNA (BRED) [10,12,13]. Using these approaches, phages have been successfully redirected from one bacterium to another, both within the genus and between genera [9,14]. Second, predictable phage design requires detailed data on the mechanisms of phage infection and replication, as well as the structural organization of the virion. Modern methods of structural biology, such as X-ray crystallography and cryo-electron microscopy, provide detailed structures of both individual proteins and whole virions; however, such methods are time-consuming, and the variety of phages is huge. The deficiency of structural data can be partially compensated using AI-based structure prediction methods such as AlphaFold 2, AlphaFold 3 [15], and similar methods [16]. As for the mechanisms of infection, many details have been revealed in recent years; however, most of the data have been obtained for popular laboratory phages, like T4, T7, and lambda [17,18,19]. Furthermore, annotated phage genomes often contain many genes whose functions are unknown, indicating an incomplete understanding of phage biology.

*Klebsiella pneumoniae* is a type of pathogenic bacterium that has developed resistance to many classes of antibiotics. This bacterium has been included in the World Health Organization’s ESKAPE group of pathogens due to its resistance and ability to cause serious infections. Therefore, the creation of therapeutic phages against *K. pneumoniae* is an important task. *Klebsiella* cells are surrounded by a protective polysaccharide capsule and more than 140 different capsule K-types have been identified [20]. To date, several studies have been published on the construction of synthetic phages that infect *Klebsiella* [9,14,21,22,23]. Some of them have shown that transplantation of genes encoding receptor-binding proteins of one phage into the genome of a related phage leads to a switch in the host range of the latter [9,14]. However, the choice of the recipient phage for gene transplantation and the impact of this choice on the biological properties and host range of the resulting phage have not been investigated properly. In this study, the genome of a related phage used to embed the gene of interest is referred to as a genomic scaffold or simply a scaffold, as proposed previously [9].

In this study, two *K. pneumoniae* lytic phages, KP192 and KP195, belonging to the *Przondovirus* genus were used to investigate the impact of genome scaffold substitution on the properties of the resulting synthetic phages. Due to differences in the tailspike proteins that are responsible for recognition and degradation of capsular polysaccharide, these phages can infect *K. pneumoniae* strains with different capsule types: KP192 infects strains of K2 and KL111 K-types, while KP195 infects K64 strains. Two synthetic phages named KP192_tspA195 and KP195_tspAB192 were constructed based on the KP192 and KP195 genomes with the swapped genes encoding tailspike proteins (*tsp* genes). This exchange can be considered both as a replacement of the *tsp* genes in the same genomic scaffold, and as a replacement of a genomic scaffold surrounding the same *tsp* gene. The capsule specificity of the synthetic phages was fully consistent with that of the tailspike proteins, as expected. However, it was found that the replication efficiency and lytic properties of the phages with the same tsp proteins but different genomic scaffolds (e.g., KP192 versus KP195_tspAB192) can differ substantially for the same *Klebsiella* strain. Moreover, for some *Klebsiella* strains, the fitness of phages with a swapped scaffold was substantially improved. This finding highlights the need to select an appropriate genomic scaffold of a related phage that provides effective counteraction to bacterial defense systems when designing therapeutically relevant synthetic phages.

## 2. Results

### 2.1. Bacteriophages KP192 and KP195 Have Different Tailspike Proteins and, Hence, Different Host Ranges

Two *Klebsiella* phages, KP192 and KP195, from the Collection of Extremophilic Microorganisms and Type Cultures (CEMTC) of the Institute of Chemical Biology and Fundamental Medicine SB RAS (ICBFM SB RAS), Novosibirsk, were previously isolated from sewage and used in this study. The KP192 and KP195 genomes are 40,635 bp and 40,540 bp, respectively (GenBank ID: NC_047968 and NC_047970). Both phages are T7-like podoviruses, and according to their genome organization and genome sequences, they are members of the *Przondovirus* genus of the *Studiervirinae* subfamily (Figure 1). Intergenomic similarity is 83% for the complete KP192 and KP195 genomes and 90% for those excluding the tailspike protein (*tsp*) genes. The KP192 genome contains two genes, *tspA192* and *tspB192*, encoding tailspike proteins tspA192 and tspB192. These proteins share 98% identity with tailspikes gp42 and gp43, respectively, from a closely related Kp9 phage [24] (GenBank ID: ON148529) (Appendix A). The Kp9_gp42 spike and, hence, tspA192 form homotrimers (PDB ID: 7Y1C and 7XY1), which is a common feature of phage tailspikes and tail fiber proteins. These proteins also contain the T7gp17-like N-terminal adapter/anchoring domain that promotes the insertion of the tailspike into the tail of the phage. In this study, the tailspikes that are directly attached to the phage tail via the T7gp17-like anchoring domain are referred to as type A tailspikes or “tailspike As”. The *tspB192* gene encodes the second type of tailspikes, referred to as “tailspike Bs” in this article, which are attached via their N-terminal conserved domain to the T4gp10-like branching domain of the tailspikes A [25,26]. As shown for the closely related phage KP32, tailspike Bs also form homotrimers [26]. The KP195 phage contains only one gene encoding tailspike A, named tspA195, which shares 97% identity (Appendix A) with a tailspike protein from the *Klebsiella* phage SH-Kp 152410 [27] (PDB ID: 8X8M; GenBank ID: NC_047908).

Since both tspA192 and tspA195 proteins contain a conserved T7gp17-like N-terminal adapter/anchoring domain, they can be substituted for each other when assembling chimeric phages with non-native tailspike proteins, as has been demonstrated previously for tailspikes from other przondoviruses [14]. AlphaFold3 modeling also confirmed that the N-terminal domains have a similar tripod joint-like fold (Figure 1C), indicating that these proteins should connect to the phage tail interchangeably. However, there are some amino acid differences within the first 116 amino acid residues of these proteins (amino acid identity is approximately 80%) that may reduce the efficiency of tail assembly.

The host strains for KP192 and KP195 phages are *K. pneumoniae* CEMTC-2274, the K-type of which has not been previously determined, and *K. pneumoniae* CEMTC-2337 with the K64 K-type, respectively. Using allele-specific primers targeting the *wzy* gene of the capsule polysaccharide synthesis locus, it was demonstrated that the CEMTC-2274 strain belongs to the KL111 K-type (Appendix A). In this study, strains CEMTC-2274 and CEMTC-2337 are referred to as A_KL111_ and E_K64_, respectively. In addition to the A_KL111_ strain, KP192 can successfully reproduce in a number of K2 capsule-type *K. pneumoniae* strains. Furthermore, neither phage KP192 nor phage KP195 can reproduce on any of the rest of the 240 *K. pneumoniae* strains from CEMTC belonging to 42 different K-types other than K2, KL111, and K64. Based on the K-types of sensitive *Klebsiella* strains and the similarity to the sequences of tailspike proteins with known specificity [24,28], the capsular specificities of proteins tspA192, tspB192, and tspA195 are KL111, K2, and K64, respectively (Figure 1 and Appendix A). Notably, the KP192 phage does not infect strains that are sensitive to the KP195 phage, and vice versa (Appendix A).

The observation that it is the tspA192 protein that allows the KP192 phage to recognize KL111-type *Klebsiella* strains indirectly confirms that other related phages (e.g., Kp9 and KpV763) that have similar tailspikes should also be able to recognize KL111 capsular-type strains. The specificity of the A tailspikes of these phages has not been previously determined, possibly because the strains used to type these phages did not include KL111 strains [24,29].

### 2.2. Design and Assembly of Synthetic Phage Genomes

In order to study the impact of genomic scaffold replacement on the properties of the synthetic phages KP195_tspAB192 and KP192_tspA195, synthetic phage genomes with swapped *tsp* genes were designed (Figure 1B). An outline of the study is shown in Figure 2A.

The genomes of all synthetic phages were assembled using transformation-associated recombination (TAR) cloning in *Saccharomyces cerevisiae* [11,30,31]. Phage genomes were divided into nine overlapping fragments of 3–6 kbp in relatively conservative regions. These fragments (Appendix A) were PCR-amplified and combined with a part of the yeast centromeric plasmid pRSII415, similar to the method described previously [9,14,32]. The detailed scheme of the assembly is shown in Figure 2B. The genomes of the parental phages KP192 and KP195 were assembled similarly to confirm that the level of mutations that may arise during assembly does not prevent the production of viable phages. These control phages were designated KP192ctrl and KP195ctrl. Following yeast transformation, individual colonies were screened using PCR. Yeast centromeric plasmid DNAs containing synthetic phage genomes were isolated from the positive clones after cultivation and were used for phage genome “rebooting”. The properties of the synthetic phages are summarized in Table 1.

### 2.3. “Rebooting” of the Klebsiella Phage Genomes Using E. coli as an Intermediate Host

Since *K. pneumoniae* strains often demonstrate low electrocompetence [33], virions of synthetic phages were produced by transformation of *E. coli* as an intermediate host [9]. *Klebsiella* RNA-polymerase promoters are recognized in *E. coli*, which ensures the expression of early *Klebsiella* phage genes, synthesis of phage proteins, and assembly of phage particles [9]. However, these phage particles cannot infect *E. coli* cells for subsequent reproduction. Therefore, *E. coli* cell extracts containing phage particles were used to infect *K. pneumoniae* strains with a suitable K-type. The *K. pneumoniae* strain A_KL111_ with the capsular type KL111 was chosen for amplification of 195_tspAB192 and KP192ctrl phages, since both had KL111-specific tspA192 tailspikes. The *K. pneumoniae* strain E_K64_ was used in the case of K64-specific phages 192_tspA195 and KP195ctrl for similar reasons.

“Rebooting” of the genomes of control synthetic phages KP192ctrl and KP195ctrl resulted in the formation of multiple plaques (Figure 2C) on *Klebsiella* lawn, and the morphology of plaques formed by control and wild-type phages was the same. This indicated a high efficiency of assembly and “rebooting” of the genomes of synthetic *Klebsiella* phages using *E. coli* as an intermediate host. The genomes of the synthetic phages 195_tspAB192 and 192_tspA195 were also successfully “rebooted”, although the number of plaques was smaller (Figure 2C).

The correct assembly of genomes of synthetic phages was confirmed using PCR verification and sequencing. First, four pairs of phage-specific primers (Appendix A) were used in PCR to verify that all the 195_tspAB192 and 192_tspA195 phages were chimeric: they contained genes from the genomic scaffold of one phage and the genes encoding the tailspike protein from another phage (Appendix A). Next, a 700 bp region covering a junction between the gene encoding the internal virion protein D from the genomic scaffold and the *tspA* gene from the donor phage was sequenced using the Sanger method. Sequencing revealed the presence of one base pair substitution in the intergenic region of one of the three clones of the KP195_tspAB192 phage. All other sequences were as expected. Finally, whole-genome sequencing was performed for one of the KP192ctrl phage clones using Illumina MiSeq. Three missense mutations and one silent mutation were found across 40,635 bp. These mutations did not interfere with phage replication.

### 2.4. The Efficiency of Phage Replication Depends on Its Genomic Scaffold and the Klebsiella Strain Used

To test whether the genomic scaffold can impact the phage properties, the infectious properties of phages KP192, 195_tspAB192, KP195, and 192_tspA195 were investigated using *Klebsiella* strains with suitable K-types. The efficiency of plating (EOP) and the efficiency of planktonic cell lysis were studied. The rate of adsorption of phages on cells, and hence the fraction of adsorbed phages, depends on the concentration of phage particles (its physical titer) [34]. Therefore, in order to correctly compare different phage samples, it was necessary to take the same number of infectious phage particles into the experiments. In this study, the equalization of phage particle concentration in phage samples was based on the fact that the structural proteins in the phage particles have a fixed copy number. Therefore, protein electrophoresis followed by densitometry was used to estimate and compare phage concentrations (Appendix A). Arbitrary units called “protein concentration-linked units” (PCLUs) were introduced for convenience throughout the study (see Materials and Methods). In addition, a pseudo-physical phage titer (titer_PP_), measured in PCLU/mL, was introduced. If samples of different phages have the same titer_PP_ values, their physical titers are also the same, even if exact concentrations (virions per ml) are unknown. This was the basis for standardization of experimental conditions in this study.

The efficiency of plating of the KP192 and 195_tspAB192 phages was studied on the A_KL111_ strain and three K2 strains of *K. pneumoniae*: CEMTC-2291, CEMTC-2573, and CEMTC-3533 (hereinafter strains B_K2_, C, and D_K2_, respectively). For this purpose, infectious titers of phage samples having the same pseudo-physical titer were determined, and their plating efficiency relative to that of phage KP192 on strain A_KL111_ (relative efficiency of plating, rEOP) was calculated (Figure 3A and Appendix A). In addition, the size and morphology of plaques were examined (Figure 3C).

Wild-type phage KP192 efficiently formed plaques (rEOP ≥ 1) using strains A_KL111_, C_K2_, and D_K2_, with plaques of medium size. However, phage KP195_tspAB192 formed smaller plaques on the same strains, and the efficiency of plating was ~10 times lower. Meanwhile, synthetic phage KP195_tspAB192 formed large plaques with a halo using the B_K2_ strain, in contrast to KP192, which formed very small plaques without a halo. Therefore, strain B_K2_ and the group of strains A_KL111_, C_K2_, and D_K2_ substantially differed in susceptibility to the studied phages, indicating that there is a correlation between some genes located within the phage genomic scaffold and the phage’s ability to replicate efficiently on particular *Klebsiella* strains.

Since EOP only partially characterizes bacteriophage fitness, the ability of phages to inhibit the growth of planktonic *Klebsiella* culture was also investigated (Figure 3E). Of the K2-strains, only B_K2_ was studied because the plaque morphology formed by the phages differed most using this strain. The efficiency of planktonic bacteria lysis was in good agreement with the plating efficiency and plaque morphology.

Similar results were observed in the case of K64-specific phages KP195 and KP192_tspA195. The efficiency of plating of these phages was studied using strain E_K64_ and three other K64 strains: CEMTC-11036, CEMTC-11038, and CEMTC-11039 (hereinafter strains F_K64_, G_K64_, and H_K64_, respectively). For the E_K64_ strain, wild-type phage KP195 had slightly higher efficiency of plating (Figure 3B and Appendix A), formed substantially larger plaques, and also lysed the planktonic culture more efficiently (Figure 3D,F). In contrast, phage KP192_tspA195 with the same spikes but the KP192 genomic scaffold replicated efficiently on strains F_K64_, G_K64_, and H_K64_. These observations clearly indicated that a match between the bacterial K-type and the phage spikes is not sufficient for efficient phage replication. In addition, some genes located in the phage’s genomic scaffold must match some genes in the bacterium.

### 2.5. Analysis of the Differences Between the Genomes of Phages KP192 and KP195 That Potentially Affect the Efficiency of Phage Reproduction

To explain the differences in phage properties caused by phage scaffold change, the genomes of phages KP192 and KP195 were compared, and the amino acid sequences of orthologous proteins were also compared (Figure 1A, Table 2). Conserved proteins (major capsid protein, portal protein, large and small subunits of terminase) differed within 3–7%. However, the differences between some proteins were approximately 20%, which can be substantial. For example, protein kinases of phages KP192 and KP195 differed by 19%, and the differences were localized within two regions of these proteins. The orthologous protein kinase of the T7 phage is known to phosphorylate a variety of host proteins, activating or inactivating their functions [35,36,37]. Nucleic acid-binding proteins including those from host anti-phage defense systems are the most phosphorylated [38]. So, it can be assumed that protein kinases of KP192 and KP195 phages perform a similar role. Hence, the differences in the two regions of these phage kinases may reflect differences in the host target proteins (presumably, belonging to anti-phage defense systems) regulated by them.

Similarly, the inhibitors of host dGTPase differed by 34%. However, the host dGTPase gene is highly conserved among different strains of *K. pneumonia*. Therefore, it is unlikely that the difference in reproduction efficiency between phages KP192 and KP195_tspAB192 was caused by differences in its dGTPase inhibitor genes. Next, the fusion proteins of phages KP192 and KP195 differed by 23%. Although genes of similar proteins were found in the genomes of many phages, no details regarding their role have been published. A blast search showed that the sequence of the N-terminal region of this protein varies greatly, even among closely related *Przondovirus* phages. Finally, the genes encoding HNH and homing endonucleases of KP195, which can be involved in phage resistance to host defense systems, were absent in the KP192 genome. Thus, this genome analysis revealed several genes that may be responsible for the difference observed in reproduction efficiency between phages KP192 and KP195_tspAB192, and between phages KP195 and KP192_tspA195.

## 3. Discussion

In order to successfully construct synthetic phages against a target bacterium, many factors should be considered. Selection of suitable phage receptor-binding proteins is necessary, since these proteins ensure the binding of the phage to the bacterial receptor and the degradation of the protective layer of the bacterium (reviewed in Dunne et al. [4] and Lenneman et al. [5]). However, replacement of receptor-binding proteins is sometimes insufficient to produce viable phages [9,14].

In this study, it was examined whether the genomic scaffold should be taken into account when designing synthetic bacteriophages with a given specificity. For this purpose, two *Klebsiella* phages from the *Przondovirus* genus were chosen as model phages. Notably, these phages, KP192 and KP195, were specific to *K. pneumoniae* strains A_KL111_ (CEMTC-2274) and E_K64_ (CEMTC-2337), respectively, each with different capsular type. The exchange of the *tsp* genes between KP192 and KP195 indicated that the resulting phages can reproduce only in strains that are recognized by the phage tailspike proteins. Therefore, the presence of an appropriate tailspike protein is necessary for successful reproduction of the phage in the *Klebsiella* strain. This result is consistent with previous studies [9,14].

However, the reproduction efficiency of synthetic phages with the same *tsp* genes but different genomic scaffolds differed substantially. This was reflected in the efficiency of plating and bacteria lysis efficiency. Thus, the parental phages, KP192 and KP195, and their synthetic analogs, KP192ctrl and KP195ctrl, were propagated more efficiently on some of the tested strains of suitable K-types than synthetic phages with swapped genomic scaffolds. However, the replication efficiency of the phages KP195_tspAB192 and KP192_tspA195 was higher compared to parental phages when using the rest of the tested strains. A similar observation was reported previously for the synthetic phage RCIP0035∆rbpIII::rbp0046 [23]. This indicates that efficient phage replication can only be achieved if some yet unidentified genes located in the phage genomic scaffold match some genes of the target bacterium.

The obtained results also indicate that target bacteria can differ in some genomic features (not related to the K-type), which can affect the efficiency of reproduction of phages based on a certain genomic scaffold. Thus, the strains A_KL111_, C_K2_, D_K2_, F_K64_, G_K64_, and H_K64_ were more suitable for the propagation of phages based on the KP192 scaffold, while the strains B_K2_ and E_K64_ were more suitable for the propagation of phages with the KP195 scaffold. Therefore, the selection of the scaffold for a synthetic phage is important, and the use of a proper one can increase lytic activity of the phage on a target group of bacterial strains.

Since a decrease in the efficiency of reproduction can significantly affect the therapeutic properties of the phage, it is necessary to select a proper genomic scaffold that would ensure effective propagation of the synthetic phage in target strains. Our findings indicate that the sequence of tailspike proteins is important but not sufficient on its own for the design of phages with predictable properties, and other factors affecting the efficiency of phage reproduction should be considered. Some genes of phages KP192 and KP195 differ significantly, and we suggest that at least some of them are involved in counteraction against host anti-phage defense systems. In addition, observed differences in phage replication efficiency may be due to the differences in more conserved genes encoding the nozzle protein, presumably involved in binding to the secondary receptor, and the gatekeeper protein, which can presumably exhibit capsule degrading activity [39,40]. Finally, the observed differences in phage replication efficiency may be related to the presence of certain sites in phage genomes that are sensitive to the action of bacterial restriction–modification systems, as has been shown for some phages that infect *Pseudomonas aeruginosa* [41]. It is not yet clear which factors are the most important for the effective reproduction of a phage in a particular target bacterium. Further accumulation of experimental data and comprehensive analysis are required to identify these factors. This, in turn, would help to transform the design of synthetic bacteriophages from a mixture of art and science into a technology.

## 4. Materials and Methods

### 4.1. Phages, Bacterial, and Yeast Strains

Two wild-type *Klebsiella* phages, KP192 and KP195 (GenBank ID: NC_047968 and NC_047970, respectively), from CEMTC of the ICBFM SB RAS, Novosibirsk, were used in this study. *Saccharomyces cerevisiae* strain BY4741 (ATCC 4040002) was applied for transformation-associated recombination (TAR) cloning. *Escherichia coli* TOP10 (Thermo Fisher Scientific, Waltham, MA, USA) was used for “rebooting” of phage genomes. *K. pneumoniae* strain CEMTC-2274 (KL111-type strain, also referred to as strain A_KL111_), K2-type strains CEMTC-2291 (strain B_K2_), CEMTC-2573 (strain C_K2_), CEMTC-3533 (strain D_K2_), and K64-type strains CEMTC-2337 (strain E_K64_), CEMTC-11036 (strain F_K64_), CEMTC-11038 (strain G_K64_), and CEMTC-11039 (strain H_K64_) were used for amplification and characterization of the phages. The K-types for these strains were determined previously using *wzi* gene sequencing (GenBank accession numbers: MN371474, MN371475, MN371483, MN371512, and MN371476, respectively) [42].

### 4.2. Culturing Conditions

Bacteria were grown at 37 °C using Lysogeny broth (LB) agar plates or in liquid LB medium with shaking (180 rpm). Suspension cultures of *S. cerevisiae* were grown at 27–30 °C with shaking (180 rpm) in rich YPD medium (1% yeast extract, 2% peptone, 2% dextrose) or selective YNB-Leu medium (0.67% Yeast Nitrogen Base with ammonium sulfate (BD), 2% dextrose, 0.069% CSM-Leu (MP Biomedicals, Santa Ana, CA, USA)).

### 4.3. Determination of the Capsular Type of the A_KL111_ Strain

The sequence of the *wzi* gene fragment of strain A_KL111_ was determined previously [42]. Using the K-PAM service [20], it was found that this *wzi* allele occurs in strains with K22, K37, and KL111 capsular types. Using the *wzy* gene sequences of these K-types, oligonucleotides were designed for genotyping (Appendix A). To determine the capsular type of A_KL111_ strain, PCR was performed followed by analysis of the products using gel electrophoresis in 1% agarose gel.

### 4.4. Preparation of DNA Fragments for Assembly of Phage Genomes

Overlapping DNA fragments of phage genomes were PCR-amplified using primers (Appendix A) and Phusion High-Fidelity DNA polymerase (Thermo Fisher Scientific, Waltham, MA, USA) according to the manufacturer’s instructions. A small drop (0.1–0.2 μL) of phage suspension (10^8^–10^10^ plaque-forming units (PFU) per ml) was used as a template. The vector fragment was amplified from the yeast centromeric plasmid pRSII-415 (Addgene #35454) using primers pRSII415_192/5_genome_dir and pRSII415_192/5_genome_rev (Appendix A) and was suitable for both KP192 and KP195 scaffold assembly. All the fragments were purified using a GeneJET Gel Extraction Kit (Thermo Fisher Scientific, Waltham, MA, USA), and DNA concentration values were determined using Nanodrop One (Thermo Fisher Scientific, Waltham, MA, USA).

### 4.5. Phage Genome Assembly in Yeast

Transformation-associated recombination (TAR) cloning in yeast was used for phage genome assembly. Competent yeast cells of strain BY4741 were prepared, as described previously [43,44]. Appropriate phage genome fragments (Appendix A) (300 ng each) along with the vector fragment (300 ng) were mixed with 240 μL of 50% PEG-3350, 36 μL of 1M lithium acetate, and 25 μL of denatured salmon sperm DNA (2 mg/mL) in a total volume of 360 μL. Approximately 10^8^ fresh competent yeast cells were added to the mixture. The cell suspension was incubated at 42 °C for 30–60 min. Yeast cells were centrifuged at 12,000× *g* for 30 s and the obtained pellet was suspended in 200 μL of sterile water. Yeast colonies were grown for 3–4 days on YNB-Leu agar plates at room temperature.

### 4.6. Yeast Colony Screening

Yeast colonies were tested for the correct assembly of the phage genome using three steps of PCR screening. At first, the presence of a junction between the phage genome region pt9 (192_pt9 or 195_pt9) (Figure 2B, Appendix A) and the vector was confirmed using primers KP192/5_40225_dir (this primer is suitable for both KP192 and KP195) and pRSII415_screening_rev, respectively (Appendix A). Second, the clones, which were positive in the first stage of PCR screening, were tested for the presence of a junction between 192_pt4 and 192_pt5 (or 195_pt4 and 195_pt5) genome regions using primers pt5_dir and pt4_rev (suitable for both phage scaffolds) (Appendix A). Finally, the presence of a junction between 192_pt1 and 192_pt2 genome regions (or 195_pt1 and 195_pt2) was confirmed using primers pt2_dir and pt1_rev. Clones, for which all three steps of PCR screening were positive, were considered positive.

### 4.7. Isolation of a Yeast Centromeric Plasmid Containing the Bacteriophage Genome

Several positive colonies were transferred into YNB-Leu medium and grown at 30 °C with shaking (180 rpm) until the optical density OD600 reached 6–9. Total yeast DNA containing a pRSII-415 plasmid bearing the phage genome was extracted from yeast cells as previously described [9,14].

### 4.8. Phage Genome “Rebooting”

Phage genomes were rebooted using *E. coli* as an initial phage propagation host, as previously described [9,14]. Briefly, electrocompetent *E. coli* TOP10 cells (Thermo Fisher Scientific, Waltham, MA, USA) were electroporated using yeast DNA sample containing the phage genome. Following electroporation, cells were grown in 1 mL of SOC medium at 37 °C for 3 h with shaking. To release phages, cells were lysed by adding 50 μL of chloroform followed by shaking and centrifugation at 12,000× *g* for 1 min. Supernatant was mixed with 0.5 mL of appropriate exponentially growing *K. pneumoniae* strain in 4 mL of molten top agar (0.8% *w*/*v*) and poured onto LB agar plates. Phage plaques were observed after incubation at 37 °C for 3–16 h.

### 4.9. Verification of Genome Assembly Accuracy and Genome Sequencing

Four pairs of phage-specific primers (Appendix A) were used in PCR to verify that the synthetic phage samples contained chimeric genomes (the genomic scaffold from one phage and the transferred tailspike gene from another phage). The primers KP192_10950bp_valid_dir and KP192_11110bp_valid_rev were used to amplify a 206 bp fragment located in the region of the KP192 phage genome between the lysozyme and primase genes. The primers KP192_tspB_36450bp_valid_dir and KP192_tspB_37050bp_valid_rev were used to amplify a 621 bp fragment located in the *tspB192* gene of phage KP192. The primers KP195_9930bp_valid_dir and KP195_10320bp_valid_rev were used to amplify a 411 bp fragment located in the gene encoding the HNH endonuclease of phage KP195 (absent in phage KP192). Finally, the primers KP195_tsp_36750_valid_dir and KP195_tsp_37000bp_valid_rev were used to amplify a 270 bp fragment located in the *tspA195* gene of phage KP195. DreamTaq DNA polymerase (Thermo Fisher Scientific, Waltham, MA, USA) was used in PCR according to the manufacturer’s instructions. A small drop (0.1–0.2 μL) of phage sample (10^8^–10^10^ plaque-forming units (PFU) per mL) was used as a template.

A 700 bp region covering a junction between genes encoding internal virion protein D and tailspike A was amplified by PCR using pt7_ivpD192/5_seq_dir and pt8_tspA192_seq_rev primers (or pt8_tsp195_seq_v2_rev, depending on the sequence of the tailspike A gene). Sanger sequencing of the fragment was performed using a BigDye Terminator v3.1 cycle sequencing kit (Thermo Fisher Scientific, Waltham, MA, USA) according to the manufacturer’s instructions.

Complete genome sequencing was performed as described previously [45]. Briefly, phage genomic DNA was fragmented using a Covaris Ultrasonicator (Covaris, Woburn, MA, USA). An NEB Next Ultra II DNA Library Prep Kit for Illumina (both from New England BioLab, Ipswich, MA, USA) was used for DNA library construction. Sequencing was performed using the MiSeq Benchtop Sequencer and MiSeq v. 2 Reagent Kit (2 × 250 base reads) (Illumina Inc., San Diego, CA, USA). The phage genome was assembled de novo using SPAdes Genome Assembler version. 3.15.4 [46].

### 4.10. Phage Propagation and Purification

Phages KP192 and KP195_tspAB192 were propagated by infecting 50 mL of an exponentially growing culture of *K. pneumoniae* strain A_KL111_ (optical density OD_600_ = 0.4–0.7) at a multiplicity of infection (MOI_inf_, i.e., the ratio of phage to bacterium, calculated based on infectious titer of the phage sample) of 0.01 (1 PFU of phage per 100 cells). Phages KP195 and KP192_tspA195 were propagated using *K. pneumoniae* strains E_K64_ or H_K64_, respectively. The infected cultures were incubated with shaking at 37 °C until bacterial lysis occurred. Bacterial debris was removed using centrifugation and phages were purified from supernatant by polyethylene glycol-6000 (PEG-6000) precipitation, as described previously [47,48]. Phage pellet was suspended in 800 μL of SM buffer (10 mM NaCl, 10 mM MgCl_2_, 50 mM Tris-HCl pH 7.5, 0.05% NaN_3_) and stored at +4 °C.

### 4.11. Determination of Infectious Titer of Phage Samples

Two types of phage titers were used in this study: infectious titer and pseudo-physical titer. To determine the infectious titer, an appropriate indicator strain was grown at 37 °C with shaking (180 rpm) until the OD_600_ reached 0.5–0.6. The culture (0.5 mL) was mixed with 4 mL of molten top agar and poured onto LB agar plates. Phage samples were diluted 10-fold in LB medium ranging from 10 ^− 1^ to 10 ^− 8^ dilution. Aliquots (6 μL) were applied to the plates. Following overnight incubation at 37 °C, phage plaques were counted. The experiments were performed in triplicate. The data were processed using Microsoft Excel 2010.

### 4.12. Determination of Pseudo-Physical Titer (Titer_PP_) of Phage Samples

In this study, the equalization (or matching) of phage particle concentrations in phage samples was based on the fact that the proteins in the phage particle have a fixed copy number. Therefore, protein electrophoresis followed by densitometry was used to estimate and compare phage concentrations. Arbitrary units called “protein concentration-linked units” (PCLUs) were introduced for convenience throughout the study. In addition, a pseudo-physical phage titer (titer_PP_), measured in PCLU/mL, was introduced. If samples of different phages have the same titer_PP_ values, then their physical titers are also the same, even if the exact number of phage particles per mL is unknown (hence, the term “pseudo” is used). The major benefit of using titer_pp_ is that it does not depend on the bacterial strain used for phage plating. In contrast, the infectious titer of a phage sample is strain-specific because it depends on a phage’s efficiency of plating on a particular strain.

The infectious titer of phage KP192 on strain A_KL111_ was chosen as a reference for determining the PCLU units in this study. Thus, if a sample of the KP192 phage had an infectious titer of 10^11^ PFU/mL when analyzed using strain A_KL111_, then it was considered to contain exactly 10^11^ PCLU in 1 mL. That is, the titer_pp_ for this sample is 10^11^ PCLU/mL. For samples of other phages, their pseudo-physical titer values were determined by electrophoresis-based comparison with the reference sample of phage KP192.

To determine the pseudo-physical titer of the phage, two-fold dilutions of the test sample were analyzed by SDS-PAGE together with a reference KP192 sample containing 3 × 10^8^ PFU (determined using the A_KL111_ strain) of PEG-purified KP192 phage. The gel (12% *w*/*v*) was stained using Coomassie G-250 (Appendix A). Densitometry analysis of bands corresponding to the major capsid protein (MW = 37 kDa) was performed using Image-Lab 6.0 software (Bio-Rad). The diluted sample whose band intensities were the most similar to the reference sample was used to calculate the pseudo-physical titer of the test phage using the following equation:titer_PP_ = OD_test_ × DF × (3 × 10^8^ PCLU)/(OD_reference_ × V_test_)(1)
where OD_test_ is the density of the major capsid protein band (approximately 37 kDa) of the diluted sample, OD_reference_ is the same parameter of the reference sample, and DF is the dilution factor. The error rate of this method is estimated as 20–30% due to the inaccuracy of gel densitometry. Method validation is described in the Appendix B section.

### 4.13. Determination of the Efficiency of Plating

Efficiency of plating reflects the probability that a single phage particle bound to a cell will lead to infection and subsequent plaque formation. In this study, the relative efficiency of plating (rEOP) of phages was used. The efficiency of plating of phage KP192 on a bacterial lawn formed by strain A_KL111_ was chosen as a reference (hence, its rEOP = 100%). The rEOP of the test phage on any test strain was calculated using the following equation:rEOP = [titer_inf_ (test)/titer_inf_ (reference)] × [titer_PP_ (reference)/titer_PP_ (test)] × 100%(2)
where titer_inf_ is the infectious titer of the test and reference samples, and titer_PP_ is the pseudo-physical titer for these samples.

Since the value of multiplicity of infection (the phage-to-cell ratio) is based on the phage titer value, two types of MOI (MOI_inf_ and MOI_PP_) can be calculated based on infectious or pseudo-physical titers, respectively. These parameters are related by the following equation:MOI_inf_ = MOI_PP_ × (rEOP/100%)(3)

In this study, the same MOI_PP_ was used to compare the properties of the phages.

### 4.14. Bacterial Killing Assay

An appropriate *K. pneumoniae* strain was cultivated at 37 °C with shaking at 180 rpm until OD_600_ reached 0.5. The test phage (10^8^ PCLU) was added to 5 mL of cells (10^9^ CFU) at MOI_PP_ = 0.1. Following 30 min of adsorption at 37 °C, the cell culture was incubated at 37 °C with shaking at 180 rpm. Starting from the moment of mixing cells with phage, 100 μL aliquots were taken and their optical density OD_600_ was measured. The experiments were performed in triplicate for each *Klebsiella* strain.

### 4.15. Bioinformatic Analysis of the Differences Between the KP192 and KP195 Genomes

Intergenomic similarity for KP192 and KP195 genomes was calculated using VIRIDIC (https://rhea.icbm.uni-oldenburg.de/viridic/ accessed on 12 September 2024) [49]. Alignment of genomes was performed using mafft v.7 (https://mafft.cbrc.jp/alignment/server/ accessed on 5 November 2024). Proteomic alignment was performed using the clinker tool (CAGECAT server, https://cagecat.bioinformatics.nl/tools/clinker accessed on 12 December 2024). Amino acid identity was calculated using the AlignX tool from Vector NTI Suite 8.0.

### 4.16. Protein Structure Modeling and Visualization

The 3D structures of the homotrimers of N-terminal domains of tailspike proteins were predicted with high fidelity using AlphaFold3 [15]. Alignment, RMSD calculation, and visualization of the models were performed using UCSF Chimera v 1.13.

### 4.17. Quantification and Statistical Analysis

Data are presented as the mean ± SD. Statistical analyses were performed using GraphPad Prism 8.0.1 (GraphPad Software). Significant differences were determined by one-way ANOVA and corrected for multiple comparisons using Tukey’s test.

## Figures and Tables

**Figure 1 ijms-26-06824-f001:**
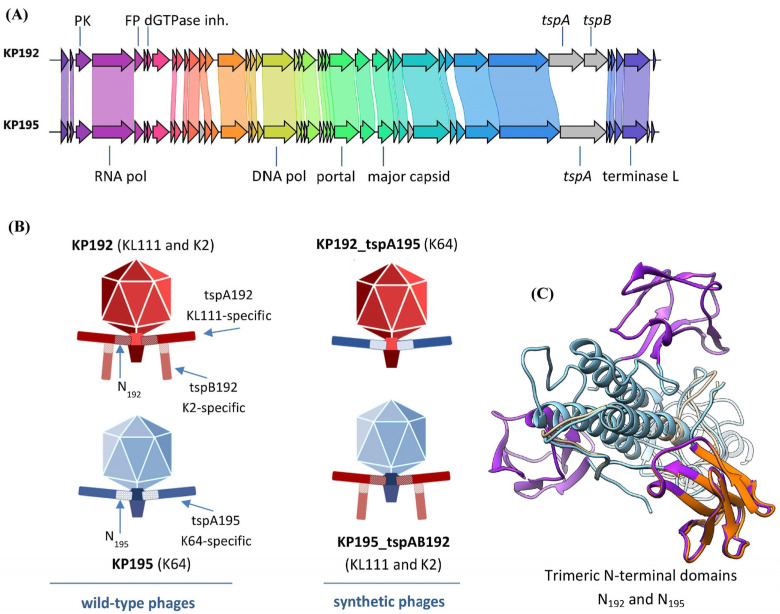
Wild-type and synthetic phages used in the study. (**A**) Alignment of KP192 and KP195 genomes. Genes sharing 90% or more sequence identity are connected by colored stripes. Genes that do not have a homologous pair in the related genome are marked in gray. “RNA pol”—RNA polymerase gene; “DNA pol”—DNA polymerase gene; “terminase L”—terminase large subunit gene; “tspA” and “tspB”—genes of tailspike proteins A and B; “PK”—protein kinase gene; “dGTPase inh.”—gene encoding the inhibitor of host dGTPase; “FP”—fusion protein gene. The image was prepared using the clinker tool. (**B**) Schematic representation of wild-type and synthetic phages. The assumed capsular specificity of synthetic phages is indicated in parentheses. “N_192_” and “N_195_”—N-terminal domains of the tspA192 and tspA195 proteins, respectively. (**C**) Aligned AlphaFold3 models (pLDDT > 0.8) of trimeric T7gp17-like N-terminal adapter/anchoring domains of tspA192 and tspA195 tailspike proteins. Three copies of the N_195_ domain are shown in light blue and purple (off-axis domains). One copy of the N_192_ domain is shown in beige and orange (off-axis domain). The C-alpha root-mean square deviation is 1.02 Å. Models are visualized using UCSF Chimera 3.13.

**Figure 2 ijms-26-06824-f002:**
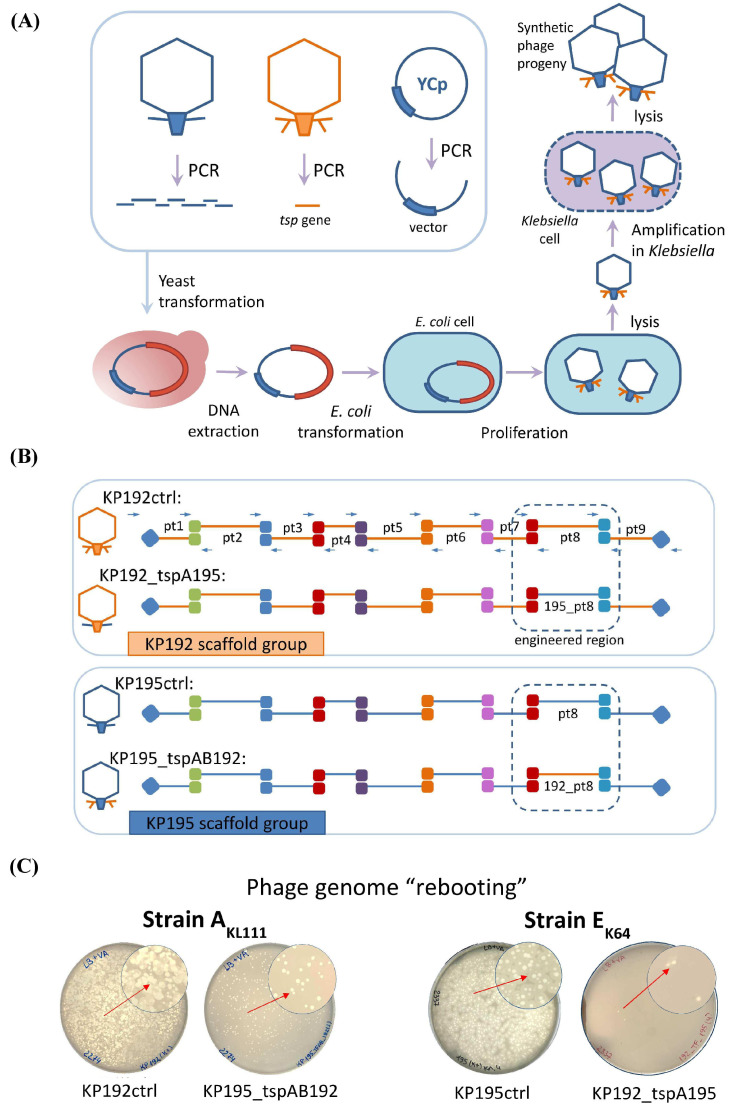
Assembly and “rebooting” of the synthetic phage genomes. (**A**) An outline of phage genome assembly in yeast and “rebooting” of phage genomes. “YCp”—yeast centromeric plasmid; “*tsp* gene”—DNA fragment containing gene(s) encoding tailspike protein(s). The red fragment represents an assembled phage genome integrated into a yeast plasmid. (**B**) Detailed diagram of the assembly of synthetic phage genomes (see also Appendix A). Primers are indicated by arrows. Regions of overlapping DNA fragments are shown as colored squares. Regions of overlap with the yeast centromeric plasmid are marked with blue diamonds. Parts 1 to 9 of the genome are designated “pt1”–“pt9”. The dashed lines indicate regions that differed between phages within the same scaffold group. (**C**) “Rebooting” of synthetic phage genomes. Plates containing *Klebsiella* lawn and phage plaques are shown.

**Figure 3 ijms-26-06824-f003:**
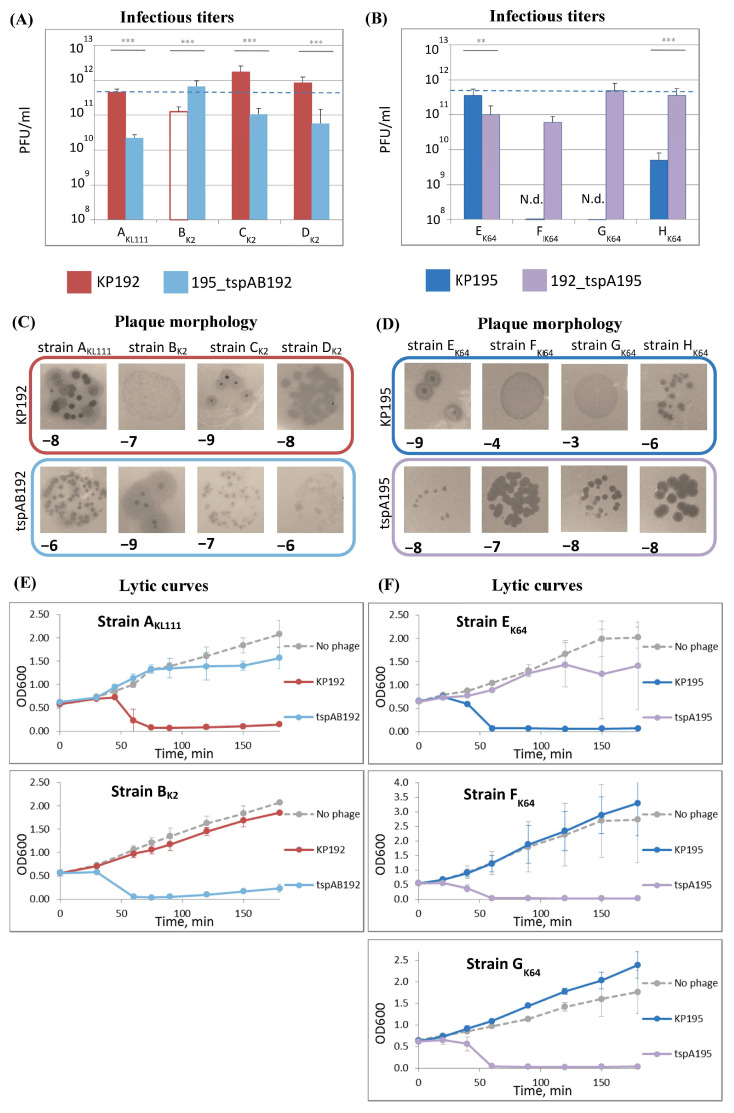
Differences in the genomic scaffolds of synthetic phages affect their replication efficiency. (**A**,**B**) Infectious titer values of phage samples for KP192 and 195_tspAB192 phages (**A**) or KP195 and 192_tspAB195 phages (**B**). Pseudo-physical titer values of all phage samples were 4.6 × 10^11^ PCLU/mL (shown as dashed line). The unpainted bar (red wireframe) for the B_K2_ strain indicates that the plaques were very small. N.d.—not determined: phage plaques could not be counted due to their small size. Data from n = 3 independent experiments are represented as mean ± SD. Statistical significance of log-transformed infectious titer values was determined using one-way ANOVA with Tukey’s multiple comparison test (** *p* < 0.01, *** *p* < 0.001). See also Appendix A for calculated relative efficiency of plating. (**C**,**D**) Plaque morphology. Numbers represent sample dilution factor, e.g., « − 8» means 1:10^8^ dilution of phage stock. Titers_PP_ of all phage samples were 4.6 × 10^11^ PCLU/mL. The scale is the same (the size of the area shown is 12 × 12 mm). (**E**,**F**) Bacteria killing (lytic) curves; 1 PCLU of phage per 10 cells was used for infection. Data from n = 3 independent experiments are represented as mean ± SD.

**Table 1 ijms-26-06824-t001:** Summary of the properties of the wild-type phages and synthetic phages constructed during the study.

	Phages with tspA192 and tspB192 Tailspikes	Phages with tspA195 Tailspikes
Name	KP192 (WT ^1^)/KP192ctrl (synthetic)	KP195_tspAB192 (synthetic)	KP195 (WT)/KP195ctrl (synthetic)	KP192_tspA195 (synthetic)
Pictogram	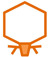	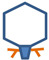	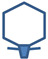	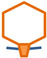
Capsular specificity	KL111 and K2	K64
Genomicscaffold	KP192	KP195	KP195	KP192
*Klebsiella* strain used for genome “rebooting”	A_KL111_	A_KL111_	E_K64_	E_K64_
Plates efficientlyon strains	A_KL111_C_K2_D_K2_	B_K2_	E_K64_	F_K64_G_K64_H_K64_
Plates poorly on strains	B_K2_	A_KL111_C_K2_D_K2_	F_K64_G_K64_H_K64_	E_K64_

^1^ Wild-type phage.

**Table 2 ijms-26-06824-t002:** Differences in orthologous proteins of KP192 and KP195 phages.

Product Name	Amino Acid Identity	Locus Tag ^1^	Note
Protein kinase	81%	HOT22_gp03, HOT24_gp04	The differences are located in two regions
Fusion protein	77%	HOT22_gp05, HOT24_gp06	The differences are located in the N-terminal region
dGTPase inhibitor	66%	HOT22_gp07, HOT24_gp08	
DNA ligase	79%	HOT22_gp08, HOT24_gp09	The differences are located in two regions
Nucleotide kinase	79%	HOT22_gp10, HOT24_gp11	
HNH endonuclease	N/A ^2^	HOT24_gp14	The gene is absent in the KP192 phage
Hypothetical protein	80%	HOT22_gp18, HOT24_gp20	
DNA polymerase	93%	HOT22_gp19, HOT24_gp21	The enzyme of the KP195 phage contains an insert near the 520 amino acid residue
Hypothetical protein	N/A	HOT24_gp24	The gene is absent in the KP192 phage
Homing endonuclease	N/A	HOT24_gp35	The gene is absent in the KP192 phage
Tailspike protein A	19%	HOT22_gp35, HOT24_gp41	
Tailspike protein B	N/A	HOT22_gp36	The gene is absent in the KP195 phage

^1^ Sequence names are given according to NC_047968 and NC_047970 GenBank records. ^2^ Not available (only one of the genomes contains the gene).

## Data Availability

All data generated in this study are available from the corresponding authors upon reasonable request.

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
