# Peer review of "Replacement of the Genomic Scaffold Improves the Replication Efficiency of Synthetic Klebsiella Phages"

_ijms, 2025, doi:10.3390/ijms26146824_

Round 1

Reviewer 1 Report

Comments and Suggestions for Authors

The manuscript describes construction of recombinant Klebsiella phages with swapped tailspike proteins, which resulted in altered host specificity. Similar approach was already used recently, but the current manuscript provides a slightly modified method and careful analyses of modified phages. The manuscript is well written and after incorporation of my suggestions I recommend it for publication.

Specific comments:

            Supplementary tables and legends to the supplementary figures were not provided with the manuscript (e.g. table of used primers). This make sometimes difficult to evaluate the results.

Line 91: Sentence: “This section may be divided by subheadings. It should provide a concise and precise description of the experimental results, their interpretation, as well as the experimental conclusions that can be drawn.” should be deleted

Line 107: structure PDB id: 7Y1C is not relevant and number should be omitted

Line 131, Fig. 1: The text does not contain information how tail fibers were attached to recombinant phage particles: did the N-terminal domain tspA195 function on KP192 phage? Is it due sequence similarity of N-terminal domains tspA192 and tspA195? (Could be mentioned in discussion.)

Line 168, Table 1: in the lane Genetic scaffold add KP195 twice for both combinations (for better clarity)

Line 199: Why recombinant phages were not whole genome sequenced? And genome sequences of host Klebsiella strains also could be useful for determination of factors influencing phage susceptibility.

Reviewer 2 Report

Comments and Suggestions for Authors

The authors are encouraged to address the comments below:

1) The current version of the Abstract is written like a part of the Introduction. The Abstract shall contain the major findings including the key methods and results. Please significantly modify the Abstract;

2) Regarding the replacement of the genomic scaffold in the Klebsiella phages, a key message needs to be explained is about the reproducibility of the assays and the stability of the synthetic Klebsiella phages. Overall, for each experiment, how many repeats/replicates were performed? For the synthetic Klebsiella phages with the replaced genomic scaffold, did you try to grow/pass the phages for certain periods/generations? 

Round 2

Reviewer 2 Report

Comments and Suggestions for Authors

Thanks for the authors' efforts in revising this manuscript.